# Targeting CA-125 Transcription by Development of a Conditionally Replicative Adenovirus for Ovarian Cancer Treatment

**DOI:** 10.3390/cancers13174265

**Published:** 2021-08-24

**Authors:** Er Yue, Guangchao Yang, Yuanfei Yao, Guangyu Wang, Atish Mohanty, Fang Fan, Ling Zhao, Yanqiao Zhang, Tamara Mirzapoiazova, Tonya C. Walser, Lorna Rodriguez-Rodriguez, Yuman Fong, Ravi Salgia, Edward Wenge Wang

**Affiliations:** 1Department of Medical Oncology and Therapeutics Research, Beckman Research Institute, City of Hope Comprehensive Cancer Center, Duarte, CA 91010, USA; eryue@coh.org (E.Y.); 601667@hrbmu.edu.cn (G.Y.); yaoyuanfei@hrbmu.edu.cn (Y.Y.); guangyuwang@hrbmu.edu.cn (G.W.); amohanty@coh.org (A.M.); zzll8107@126.com (L.Z.); tmirzapoiazova@coh.org (T.M.); twalser@coh.org (T.C.W.); rsalgia@coh.org (R.S.); 2The First Affiliated Hospital, Harbin Medical University, Harbin 150001, China; 3The Cancer Hospital, Harbin Medical University, Harbin 150081, China; yanqiaozhang@ems.hrbmu.edu.cn; 4Department of Pathology, Beckman Research Institute, City of Hope Comprehensive Cancer Center, Duarte, CA 91010, USA; ffan@coh.org; 5Department of Surgery, Beckman Research Institute, City of Hope Comprehensive Cancer Center, Duarte, CA 91010, USA; lorrodriguez@coh.org (L.R.-R.); yfong@coh.org (Y.F.)

**Keywords:** conditionally replicative oncolytic adenovirus, ovarian cancer, CA-125, *MUC16*, transcription, targeted therapy, tumor specific promoter

## Abstract

**Simple Summary:**

Ovarian cancer is the fifth most common cause of cancer-related death among women in the US, and new treatments are urgently needed to help those suffering from this deadly disease. A unique feature of ovarian cancer is that a protein called CA-125, encoded by a gene called *MUC16*, is highly expressed in more than 80% of patients. To date, targeting *MUC16*/CA-125 for ovarian cancer treatment has not been successful. Here, we describe the development of an artificial virus, called a conditionally replicative oncolytic adenovirus (CRAd), that can only grow in and destroy cancer cells that express CA-125, but not normal cells. We document promising anti-cancer effects of the virus in both human ovarian cancer cells and in animal cancer models. Collectively, our study findings suggest that the development of CRAd targeting *MUC16*/CA-125 represents a unique and practical approach to ovarian cancer treatment.

**Abstract:**

CA-125, encoded by the *MUC16* gene, is highly expressed in most ovarian cancer cells and thus serves as a tumor marker for monitoring disease progression or treatment response in ovarian cancer patients. However, targeting *MUC16*/CA-125 for ovarian cancer treatment has not been successful to date. In the current study, we performed multiple steps of high-fidelity PCR and obtained a 5 kb DNA fragment upstream of the human *MUC16* gene. Reporter assays indicate that this DNA fragment possesses transactivation activity in CA-125-high cancer cells, but not in CA-125-low cancer cells, indicating that the DNA fragment contains the transactivation region that controls specific expression of the *MUC16* gene in ovarian cancer cells. We further refined the promoter and found a 1040 bp fragment with similar transcriptional activity and specificity. We used this refined *MUC16* promoter to replace the E1A promoter in the adenovirus type 5 genome DNA, where E1A is an essential gene for adenovirus replication. We then generated a conditionally replicative oncolytic adenovirus (CRAd) that replicates in and lyses CA-125-high cancer cells, but not CA-125-low or -negative cancer cells. In vivo studies showed that intraperitoneal virus injection prolonged the survival of NSG mice inoculated intraperitoneally (ip) with selected ovarian cancer cell lines. Furthermore, the CRAd replicates in and lyses primary ovarian cancer cells, but not normal cells, collected from ovarian cancer patients. Collectively, these data indicate that targeting *MUC16* transactivation utilizing CRAd is a feasible approach for ovarian cancer treatment that warrants further investigation.

## 1. Introduction

Ovarian cancer, the umbrella term for ovarian epithelial cancer, fallopian tube cancer, and primary peritoneal cancer [1], is the fifth most common cause of cancer-related death in US women [2]. In 2021 alone, 21,410 new cases of ovarian cancer were estimated, and 13,770 US women died from this cancer (American Cancer Society, Cancer Facts and Figures, 2021). As there is no effective screening test, ovarian cancer patients typically present with later-stage tumors (FIGO stage III or IV) [2], and one-third have clinically apparent malignant ascites when the primary tumor is first detected [3]. At present, the most common treatment plan for ovarian cancer patients is surgery plus chemotherapy, but the five-year survival rate for patients utilizing these approaches is less than 50% (SEER Cancer Statistics Review). Thus, there is a critical unmet clinical need for alternative treatment strategies for patients suffering from this deadly disease.

A unique feature of ovarian cancer is that more than 80% of patients express a high serum level of CA-125, one of the largest, hyperglycosylated, human proteins expressed in and shed from ovarian cancer cells. CA-125 levels in ovarian cancer patients can reach up to hundreds and thousands of units per milliliter (U/mL), while most other cancer patients and individuals without cancer have a low level of CA-125, below 35 U/mL or undetectable [2,4,5]. Bioinformatic analysis indicates that CA-125 mRNA is highly expressed in ovarian cancer cells (TCGA database, Figure 1A), but not in most other cancer cells or in normal cells.

Since its discovery in 1981 [6], CA-125 has been regarded as an ideal target for ovarian cancer treatment [7]; however, targeting CA-125 for treatment has not been successful to date. CA-125-targeted antibodies, conjugated with or without radioactive isotopes or cytotoxic drugs, such as Oregovomab and DMUC5754A, have been developed but provided no clinical advantage in large randomized placebo-controlled trials [8,9,10,11]; although, robust immune responses have been observed [12]. Adoptive T cell therapy using CA-125-directed chimeric antigen receptors has also been developed for ovarian cancer [13], and a phase I trial has been proposed [14], but no further information or clinical benefit has been reported.

The gene that encodes CA-125, termed *MUC16*, is located on chromosome 19p13.2 and is comprised of approximately 179 kb of genomic DNA [7,15]. An explanation for why CA-125 is highly expressed in gynecological cancer cells, especially in ovarian cancer cells, but rarely in other cancer cells remains elusive. Transcriptional regulation of *MUC16* is also still poorly defined [16,17]. Thapi et al. initially investigated the *MUC16* promoter and planned to assess the putative transcriptional factor that controlled CA-125 expression, as presented in an AACR abstract in 2011 [16], but no follow-up information has been published. Likewise, Zhang et al. screened for possible *MUC16* promoters and attempted specific expression utilizing follicle-stimulating hormone peptide-conjugated gro-α shRNA nanoparticle delivery to ovarian cancer cells [17].

Here, we advance the field via identification of the transactivation sequence of human *MUC16*, with which we replaced the E1A promoter in the adenovirus type 5 (Ad5) genome that controls expression of E1A, and successful development of a conditionally replicative adenovirus (CRAd), Ad5/MUC16-1040/TK-EGFP, which can only replicate in and lyse cancer cells expressing CA-125. Based on our study findings, targeting *MUC16* transactivation utilizing CRAd may now represent a feasible approach for ovarian cancer treatment.

## 2. Materials and Methods

### 2.1. Cell Lines and Culture

Ovarian cancer cell lines, CAOV3, HEYA8, OVCAR4, OVCAR8, PEO4 and TOV-112D, a mouse ovarian cancer cell line, ID8, a human embryonic kidney cell line, HEK 293, a human cervical cancer cell line, HeLa, and a human fibroblast cell line derived from ovarian cancer tissue, OVFB-1, were cultured in Dulbecco′s Modified Eagle Medium (DMEM) (Corning, Manassas, VA, USA), supplemented with 10% fetal bovine serum (FBS) (Biowest, Riverside, MO, USA) and 1× penicillin-streptomycin (Corning), in a humidified incubator (5% CO_2_) at 37 °C. Other ovarian cancer cell lines, A2780, IGROV-1, Kuramochi, OVCAR3, OVCAR5, and SKOV3, a human lung adenocarcinoma cell line, A549, and a human immortalized bronchial epithelial cell line, BEAS-2B, were cultured in RPMI 1640 (Corning), supplemented with 10% fetal bovine serum (FBS) and 1× penicillin-streptomycin, in a humidified incubator (5% CO_2_) at 37 °C.

### 2.2. Cell Viability Assay

ID8 cells labeled with a stable firefly luciferase gene were plated in 96-well plates (5000 cell/well) and cultured overnight. Ad5/MUC16-1040/TK-EGFP were added at a multiplicity of infection (MOI) of 20:1. Ganciclovir (GCV; Sigma-Aldrich, St. Louis, MO, USA) was then added to the cells at a final concentration of 10 μM and incubated for an additional 3 days. D-luciferin (potassium salt; Syd Labs, Hopkinton, MA, USA) was added to each well at a final concentration of 100 μg/mL. Luminescence intensity was measured by a Tecan Spark 10M multimode microplate reader.

Alternatively, we used the Coomassie blue stain to visualize live cells attached to the culturing plate after virus infection. Infected and dead cells were floated and washed out with PBS. The attached live cells were stained with 0.1% Coomassie Brilliant Blue R-250 (Research Products International, Mount Prospect, IL, USA) in 50% methanol and 10% glacial acetic acid for 10 min. The plate wells were washed three times with tap water and dried for image scanning.

### 2.3. Primary Ovarian Cancer Cell Collection and Culture

Primary ovarian cancer cell collection from patient ascites or pleural effusion was approved by the City of Hope Institutional Review Board (IRB; #07047). Patients were consented prior to fluid collection per the IRB-approved protocol. Laboratory use of the de-identified primary cells was approved under COH IRB #17478.

A portion of the collected fluid was sent for cytology evaluation, including examining the direct smear, cytospin, and cellblock slides using Papanicolaou stain, Diff-Quik stain, and H&E stain, per standard methods. Immunohistochemistry for expression of CA-125 (Clone OC125, Cell Marque, Rocklin, CA, USA), p53 (Clone BP53-11, Ventana, Oro Valley, AZ, USA), PAX8 (Clone MRQ-50, Ventana), WT-1 (Clone 6F-H2, Ventana), and calretinin (Clone SP65, Ventana) was performed to confirm the diagnosis.

Live cells in the collected fluid were spun down and washed once by PBS. Red cells were removed by ammonium chloride buffer. Nucleated cells, including cancer cells (isolated single cells or cell clusters), immune cells, epithelial cells, mesothelial cells, and other cell types were suspended in DMEM with high glucose, supplemented with 10% FBS, 20 mM L-glutamine, 1× insulin-transferrin-selenium solution (GenDEPOT, Katy, TX, USA), and 1× penicillin-streptomycin, seeded in tissue culture dishes or plates, and cultured in a humidified incubator with 5% CO_2_ at 37 °C. The next day, floating cells were washed out gently, and CRAd was added to the cells at an MOI of 100:1. Infected cells were observed under an inverted phase-contrast laser microscope, and images were captured with the integrated digital camera.

### 2.4. Adenovirus Construction, Amplification and Purification

The AdenoQuick 2.0 system (OD260 Inc., Boise, ID, USA), which consists of four shuttle vectors (pAd1127, pAd1128, pAd1129, and pAd1130), was used for adenovirus engineering. Each vector contains part of the wild-type human Ad5 genome that can be genetically modified and recombined for adenovirus packaging in HEK293 cells. pAd1127 contains the left end of the Ad5 genome with an E1A expression cassette. We replaced the E1A promoter with the 1040 bp upstream DNA fragment of *MUC16* with promoter activity in pAd1127, where E1A is an essential gene for adenovirus replication [18,19,20]. In the E3 region of pAd1129, we inserted a fusion gene, TK-EGFP, which encodes a fusion protein of EGFP and herpes simplex virus-1 thymidine kinase (HSV-TK). The construct sequences were verified by restriction digestion and sequencing, respectively. The adenovirus genome fragments from the four vectors were collected and ligated to generate full-length adenovirus genomic DNA with the inserts of interest, which was subsequently packaged into lambda phage heads for amplification in *E. coli*. The recombined adenovirus genome DNA was exercised and transfected to HEK293 cells for packaging, per the manufacturer’s instructions. Recombined adenoviruses were amplified in HEK293 cells and purified by cesium chloride ultracentrifugation. Purified viruses were stored in GTS buffer (2.5% glycerol, 25 mM NaCl, and 20 mM Tris-HCl, pH 8.0), per the manufacturer’s instructions.

### 2.5. Reporter Vector Construction and Dual-Luciferase Assay

The homo sapiens mucin 16 (cell surface-associated, *MUC16*) gene sequence was retrieved from GenBank (NG_055257). Upstream DNA fragments of *MUC16* were amplified by high-fidelity PCR and cloned into a reporter vector, pGL4.14 (Promega, Madison, WI, USA), to control the expression of the built-in firefly luciferase gene. Reporter plasmids were transfected into the appropriate cultured cells, and transcriptional activity was measured using the Dual-Luciferase^®^ Reporter Assay System (Promega), per the manufacturer’s instructions.

### 2.6. mRNA Isolation and Quantitative Real Time PCR (qRT-PCR)

Total RNA was extracted using the Qiagen RNeasy Mini Kit (Qiagen, Germantown MD, USA), and complementary DNA was synthesized using the Quantabio qScript cDNA SuperMix Kit (Qiagen). Real-time PCR was performed using the Applied Biosystems’ Power SYBR Green PCR Master Mix (Applied Biosystems, Warrington, UK) and ABI Prism 7900HT Sequence Detection System (Applied Biosystems, Warrington, UK). The *MUC16* primers were as follows: (sense) 5′-ACAAACTAGCAAAATAGGCTGT-3′ and (antisense) 5′-CTTCTTAATGTTTTTGGCATCT-3′. The threshold cycle number (Ct) for gene expression was calculated, and GAPDH was used as an internal control as follows: (sense) 5′-AAGAAGGTGGTGAAGCAGGC-3′ and (antisense) 5′-TCCACCACCCTGTTGCTGTA-3′. The relative fold induction was derived from the ΔΔCt values for gene expression.

### 2.7. Virus Oncolytic Activity in NSG Tumor Xenograft Models

All animal experiments were approved by the City of Hope Institution Animal Care and Use Committee (IACUC, #18013). Per the IACUC-approved protocol, female NSG mice (6–8 weeks old) were purchased from the Jackson Laboratory (Sacramento, CA, USA) and acclimated for 1 week before being started on any study. Kuramochi/FL cells (5 × 10^6^) in 100 µL of PBS were injected intraperitoneally (ip) into mice on day 0. Ad5/MUC16-1040/TK-EGFP was delivered at a dose of 10^9^ pfu/mouse on day 1, as was the vehicle (buffer) control. For the OVCAR4 model, 3 × 10^6^ cells in 100 µL of PBS were injected ip on day 0, the virus was delivered ip at 10^9^ pfu/mouse on day 8, as was the vehicle (buffer) control. An additional group was included in the OVCAR4 model; specifically, we treated the OVCAR4 tumor-bearing mice with an ip injection of 3 × 10^5^ OVCAR4 cells infected with the virus at an MOI of 100:1 on day 8. Mice were monitored closely and euthanized when cancer-related symptoms developed, such as weight loss, dehydration, abdomen distention, and impaired ambulation, per the IACUC-approved protocol.

### 2.8. Bioluminescent Imaging of Tumor-Bearing Mice

Mice bearing the ovarian cancer cells labeled with firefly luciferase were imaged at indicated timepoints utilizing Spectral Instruments’ SPECTRAL Imaging System (Lago X, Tucson, AZ, USA). Specifically, each mouse received an intraperitoneal injection of 2 mg D-luciferin in 0.1 mL of deionized water, bioluminescent images were captured, and intensity was quantified using Spectral Instruments’ companion Aura Imaging Software 4.0.

### 2.9. Statistical Analysis

Data collected from in vitro and in vivo experiments were analyzed by standard statistical methods, including student’s *t*-test, ANOVA test, and Kaplan-Meier survival analysis using GraphPad Prism 8.0.

## 3. Results

### 3.1. Identification of an Upstream Region of the MUC16 Gene with Specific Transcriptional Activity

We retrieved the DNA sequence of homo sapiens *MUC16* located on chromosome 19p13.2 (NG_055257) from GenBank. We performed multiple steps of high-fidelity PCR on genomic DNA isolated from the OVCAR3 ovarian cancer cell line and obtained a 5 kb DNA fragment (chr19:8980639-8985671). This 5 kb DNA fragment included a 4.5 kb region upstream from the transcription start site (TSS) and 0.4 kb region downstream from the TSS, and it contained both the 5’ untranslated region (5’UTR) and the first 136 bp of the open reading frame (ORF). We cloned this DNA fragment into the multi-cloning site of a firefly luciferase reporter vector, pGL4.14. We tested for transcriptional activity in HEK293 and HeLa cells because CA-125 mRNA is reportedly low (0 NX) in HEK293 cells and high (27.6 NX) in HeLa cells [21]. Both HEK293 and HeLa are rapidly growing cells that are easy to transfect with plasmid DNA in vitro. As shown in Figure 1B, the 5 kb fragment showed higher transcriptional activity in HeLa cells than in HEK293 cells. We further refined the sequence and arrived at a 1040 bp fragment (chr19: 8981294-8982333) that retains this differential transcriptional activity; 7.0 times higher activity in HeLa cells than in HEK293 cells. We next compared the *MUC16*-1040 fragment with the Ad5/E1A promoter. As shown in Figure 1C, the *MUC16*-1040 fragment showed equivalent or slightly higher activity in HeLa cells compared to the E1A promoter, but much lower activity (27.3%) in the A549 lung adenocarcinoma cell line, which is characterized by ease of use for adenovirus amplification and low expression of CA-125. Identification of a *MUC16* fragment around 1kb or less with promoter activity that specifically controls CA-125 expression makes it possible for us to engineer the adenovirus type 5 for targeted cancer therapy.

### 3.2. Generation of a CRAd with MUC16 Fragment with Promoter Activity to Control E1A Expression

We engineered the Ad5 genome DNA using the *MUC16*/1040-bp fragment to replace the E1A promoter to control E1A expression, as illustrated in Figure 2A. In the E3 region, we inserted a fusion gene, TK-EGFP, with both EGFP activity that can be used to track virus infection and herpes simplex virus-1 thymidine kinase (HSV-TK) activity that can convert pro-drug, ganciclovir (GCV), or other analogs, into a toxic metabolite to kill host cells [22,23]. We purposely incorporated the HSV-TK gene for two reasons. First, we can deliver GCV to kill infected cells if viral replication is limited and insufficient to destroy cells. Second, if viral replication is out of control, we can use pro-drug to kill the host cells before mature viruses are generated and infect other cells, thereby limiting cytotoxic effects. We transfected the recombined Ad5 genome into HEK293 cells, a human embryonic kidney cell line immortalized by Ad5 E1A/B that is routinely used for adenovirus packaging and amplification, because it expresses abundant E1A protein in the cells [24]. In the transfected HEK293 cells, we have successfully generated a recombined adenovirus that we termed Ad5/MUC16-1040/TK-EGFP (Figure 2B). This virus induces classic cytopathic effects (CPE) in CA-125-high cells, including HeLa (Figure 2C) and OVCAR3 (Figure 2D). We further tested the sensitivity of the virus-infected cells to GCV. As shown in Figure 2E, mouse ovarian cancer cell line ID8 became more sensitive to GCV when infected with the virus.

### 3.3. Replication and Oncolysis of Ad5/MUC16-1040 in Ovarian Cancer Cell Lines Is Dependent on CA-125 Expression

We tested Ad5/MUC16-1040/TK-EGFP in a panel of cell lines, including HeLa, HEK293, A2780, A549, BEAS-2B, CAOV3, HeyA8, Kuramochi, OVFB-1, OVCAR3, OVCAR4, OVCAR5, OVCAR8, PEO4, and SKOV3 for replication and oncolysis to determine if it is correlated with CA-125 expression in these cells. The CA-125 mRNA level in HEK293 cells is low [21], so it served as a comparator for the mRNA levels in other cell lines. As shown in Figure 3A, HeLa cells expressed a relatively high level of CA-125, 26.1 times higher than HEK293 cells, which is consistent with the Protein Atlas database [21]. OVCAR3, OVCAR4, CAOV3, and Kuramochi expressed a high level of CA-125 mRNA, with OVCAR3 having the highest, 145.8 times higher than HEK293 cells. Other cells, including A2780, OVCAR5, OVCAR8, PEO4, and SKOV3, expressed low levels of CA-125 mRNA. We found that the virus does replicate in and lyse CA-125 high cancer cells, including HeLa, OVCAR3 (Figure 2D and Figure 3B), OVCAR4 (Figure 3B), CAOV3, and Kuramochi cancer cells. Amplified viruses released from the cancer cells upon cell lysis can subsequently infect and kill other cells. The virus does not replicate well in the CA-125 low or negative cancer cell lines A2780, OVCAR5, SKOV3, or PEO4 (Figure 3A,B). Thus, it appears that oncolysis mediated by the virus correlates to CA-125 mRNA expression in this panel of cells. Of importance, the ovarian cancer cells lines, A2780 [25,26], OVCAR5 [19], and PEO4 [27] express coxsackievirus and adenovirus receptor (CAR) on their cell surface, and they are permissive to wild-type adenovirus replication; however, these cells are resistant to our virus, and no obvious CPE was observed even when infected at an MOI of 100:1. Both OVCAR3 and SKOV3 have limited CAR expression, but it is sufficient for wild-type adenovirus to infect and replicate in these two cell lines [28]. Our virus can infect, replicate, and cause CPE in OVCAR3 cells, but not in SKOV3 cells, which we believe is due to high expression of CA-125 in OVCAR3 cells and low expression in SKOV3. We also determined E1A expression levels via Western blot in selected ovarian cancer cell lines infected with Ad5/MUC16-1040/TK-EGFP. As shown in Figure 3C, E1A expression is high in the virus-infected OVCAR3 and OVCAR4, but low or undetectable in A2780, OVCAR5, OVCAR8, PEO4, and SKOV3. Full Western Blots are available in Appendix A.

Further, we collected normal cell lines, including the HFF human foreskin fibroblasts, the human lung epithelial cell line BEAS-2B, and two human fibroblast cell lines derived from ovarian tumor tissue established in our lab, and tested our virus on them. As expected, Ad5/MUC16-1040/TK-EGFP does not induce apparent CPE in these normal cells at an MOI up to 100:1 after a period of 2 weeks in culture with regular passaging every 3–7 days, although we can see EGFP signal 3–5 days after infection, which gradually disappears with continued cell passaging. Collectively, these results indicate that our virus can replicate in and lyse cancer cells expressing CA-125, not in cancer cells without CA-125 expression or in normal cell lines.

### 3.4. Selective Replication and Oncolysis in Primary Ovarian Cancer Cells

For translational purposes, we tested the Ad5/MUC16-1040/TK-EGFP on primary ovarian cancer cells collected from two patients; case 1 was from pleural effusion, and case 2 was from ascites. Cytology confirmed the presence of malignant cells positive for CA-125 in both cases, as shown in Figure 4 (A: case 1, B: case 2). Immunohistochemistry showed that the malignant cells were positive for PAX8 (Figure 4A,B) and WT1 (Figure 4A), negative for the mesothelial marker, calretinin (Figure 4A), p53 mutation pattern (Figure 4A, abnormal strong diffuse expression), and p53 null pattern (Figure 4B), consistent with a high-grade serous carcinoma of ovarian primary origin of both cases.

We infected the collected cells with Ad5/MUC16-1040/TK-EGFP. Five days after infection, we saw EGFP signals in cancer cells, which is easy to recognize when clustered together (Figure 4A, case 1, day 5) under an inverted phase-contrast microscope. A few more days later, the EGFP positive cells showed classic CPE, floated, and eventually died as shown in Figure 4A (day 14). Surrounding cells, mostly fibroblasts, continued to grow and replaced the culture surface where cancer cells had been attaching. Case 2 has a large number of malignant cells that are sensitive to the virus infection indicated by EGFP signal as shown in Figure 4B (case 2, middle panel, day 5), with limited normal cells in culture. A few more days later, cancer cells floated and disappeared gradually, while fibroblasts took place of the culture surface (Figure 4B, lower panel, day 14). Our preliminary data indicates that Ad5/MUC16-1040/TK-EGFP can indeed replicate in and lyse primary cancer cells collected from patients, but not in normal cells as represented by fibroblasts.

### 3.5. Oncolytic Activity in Ovarian Cancer Xenograft Models

To assess oncolytic activity in vivo, we tested Ad5/MUC16-1040/TK-EGFP in an ovarian cancer xenograft mouse model using NOD-*scid* IL2Rgamma^null^ (NSG) immunodeficient mice. Kuramochi human ovarian cancer cells labeled with a firefly luciferase gene were inoculated ip on day 0, and the Ad5/MUC16-1040/TK-EGFP virus was injected ip on day 1. As shown in Figure 5A, B, firefly luciferase intensity increases over time in the control group; However, the intensity of firefly luciferase in mice treated with ip injection of Ad5/MUC16-1040/TK-EGFP is significantly lower than the control group on day 50. NSG mice treated with ip injection of Ad5/MUC16-1040/TK-EGFP virus survived longer than the control group (Figure 5C). OVCAR4 cells were also inoculated ip on day 0, and the Ad5/MUC16-1040/TK-EGFP virus was injected ip on day 8. As shown in Figure 5D, NSG mice treated with an intraperitoneal injection of Ad5/MUC16-1040/TK-EGFP virus survived longer than the control group.

As the adenovirus particle is extremely small with a diameter of about 80–120 nm and the peritoneum cavity is large, especially in humans, virus injected directly into the peritoneum may disappear quickly, limiting cancer cell infection efficiency [29]. Thus, in the OVCAR4 model, we also tested whether virus-infected cells could carry and amplify the virus in the peritoneum and release the virus upon maturation to infect other cancer cells. As shown in Figure 5D, intraperitoneal delivery of the virus by Ad5/MUC16-1040/TK-EGFP-infected OVCAR4 cells used as a carrier also suppressed tumor cell growth and prolonged mouse survival. At the time of manuscript submission, corresponding to 166 days post-injection, two mice in the cohort remained alive without any evidence of cancer growth.

## 4. Discussion

Success in developing a CRAd that can only replicate in and lyse cancer cells expressing CA-125 provides proof-of-concept that transactivation of *MUC16*/CA-125 can be targeted for ovarian cancer treatment. Further refining the transactivation elements and optimizing the oncolytic activity may yield a potent targeted agent that can be translated from bench to bedside. More detailed future investigations of the mechanism underlying transactivation of *MUC16*/CA-125 in ovarian cancer cells may yield additional novel targets for ovarian cancer treatment.

Among all the oncolytic viruses, we chose adenovirus as our platform for hypothesis testing for the following reasons: First, adenovirus is one of the most common viruses, and it has been well characterized, demonstrating a relatively safe toxicity profile and a broad spectrum of targeted cells, including cancer cells and normal cells. Second, it is a nonenveloped virus with an icosahedral nucleocapsid containing a double-stranded DNA genome, which is highly immunogenic and induces potent and sustained T and B cell responses. Recently, adenovirus has served as an ideal vector for vaccine development [30], including many of the COVID-19 vaccines [31]. Third, human adenovirus can cross species, infect, and replicate in mouse cancer cells [32], although to a lesser degree, a feature allows us to investigate oncolytic virus-induced immune responses in immune-competent mice.

Oncolytic viruses eradicate cancer cells by two major distinct mechanisms: oncolytic activity that directly lyses infected cells, and anti-cancer immune responses that are indirectly induced by virus infection of cancer cells. To date, none of the available oncolytic viruses, FDA-approved [33] or those under investigation [34,35,36,37], have been potent enough to eradicate all cancer cells in animal models or cancer patients. Enhancement of oncolytic activity is one of the main future directions in the field, however, off-target toxicity remains a big concern. The development of oncolytic viruses with selective replication capacity in cancer cells, not in normal cells, is reassuring. Our 1040-bp *MUC16* transactivation sequence shows similar or higher transcriptional activity compared to the E1A promoter in HeLa cells, a cervical cancer cell line with a relatively high CA-125 level, but limited activity in A549 cells, a lung adenocarcinoma cell line frequently used for adenovirus amplification but with a relatively low CA-125 level. We anticipated that the replication capacity of the Ad5/MUC16-1040 series would be equivalent to the wild-type adenovirus in CA-125-high cells and would be limited or null in CA-125-low cells, thus providing an overall favorable toxicity profile.

Adenovirus is highly immunogenic and can induce overwhelming cellular responses, such as HLA expression, antigen presentation, the release of cytokines, and inflammatory changes in the infected cells and surrounding tissues, that collectively stimulate an anti-virus immune response [38]. It may also simultaneously induce an anti-cancer immune response [39] simutanously. Further modification of the virus via insertion of immunoregulatory genes into the E3 region [40], which is not essential for viral replication [41,42], may enhance or amplify this anti-cancer immune response. Because our CARd selectively replicates in CA-125-high cancer cells, not in normal cells, it may outperform other oncolytic adenoviruses for its ability to induce a potent and targeted anti-cancer immune response.

Of interest, injection of virus-infected cells generally yielded a better outcome with prolonged survival than direct virus particle injection (Figure 5D). One explanation may be that the infected cells are retained in the abdomen as virus carriers and held adjacent to the pre-inoculated cancer cells. Upon lysis, viruses may then be released to infect the adjacent cancer cells. Conversely, directly injected viruses may be rapidly cleared within a few minutes [29], possibly by diffusion to circulation, neutralization by pre-existed antibodies, or absorption by cells lining the peritoneum. It is practical to obtain cancer cells from ovarian cancer patients with ascites, infect the collected cells with the virus, and return the infected cells back to the abdomen as virus carriers. Alternatively, the infected cancer cells may serve as a whole-cell vaccine that can be used to stimulate an anti-cancer immune response to eradicate other cancer cells.

Adenovirus, either replication-competent or -deficient, is relatively safe and well-tolerated in experimental animal models and clinical trials [43,44]. However, hepatotoxicity is one potentially life-threatening reaction to adenovirus delivery [44,45]. In our studies, we did not observe hepatotoxicity or any other noticeable side effects when mice were injected intraperitoneally at a dose of 10^9^ pfu/mouse. As our virus can only replicate in CA-125-expressing cells, it is expected to have a more favorable toxicity profile that may facilitate its translation from the laboratory into clinical practice.

## 5. Conclusions

We have successfully developed a CRAd that selectively replicates in and lyses ovarian cancer cells expressing CA-125, but not normal cells. Our data indicate that targeting *MUC16* transactivation for ovarian cancer treatment by conditionally replicative oncolytic virus development is practical and warrants further investigation.

## Figures and Tables

**Figure 1 cancers-13-04265-f001:**
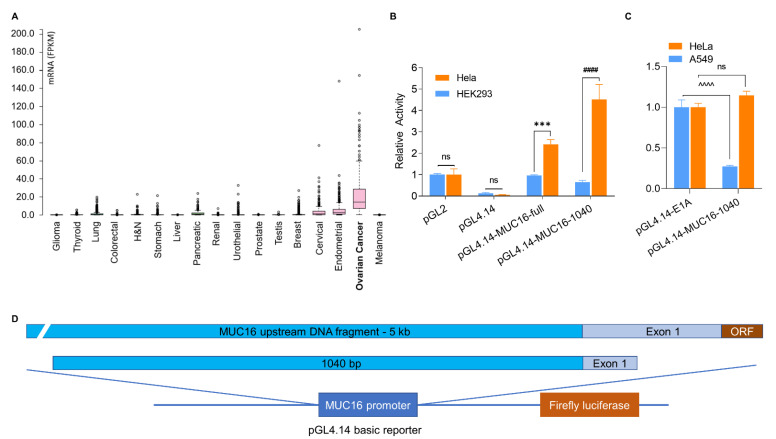
Identification of specific transcription of *MUC16*. (**A**). CA-125 mRNA expression in human cancers (TCGA database). (**B**). Relative activity of *MUC16* upstream fragment in HEK293 cells (CA-125-low) and HeLa cells (CA-125-high), compared to the pGL-2 reporter. (**C**). Relative activity of *MUC16* fragments in A549 cells (CA-125-low) and HeLa cells, compared to the Ad5/E1A promoter. (**D**). Illustration of *MUC16* upstream fragments and construction of firefly luciferase reporter. *** *p* = 0.0002 pGL4.14-MUC16-full: HeLa vs. pGL4.14-MUC16-full: HEK293, ^####^ *p* < 0.0001 pGL4.14-MUC16-1040: HeLa vs. pGL4.14-MUC16-1040: HEK293, ^^^^ *p* < 0.0001 pGL4.14-E1A: A549 vs. pGL4.14-MUC16-1040: A549, ns, no statistical significance, *n* = 3.

**Figure 2 cancers-13-04265-f002:**
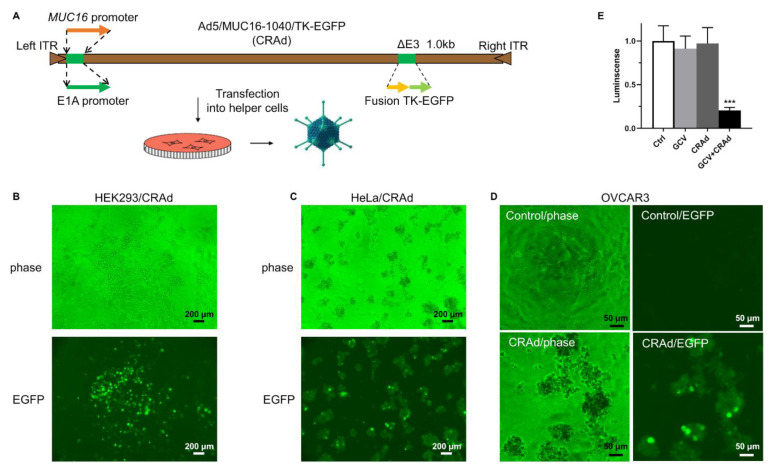
Generation of recombinant CRAd with *MUC16* 1040 bp upstream fragment to control E1A expression. (**A**). Illustration of CRAd genome recombination. E1A promoter of Ad5 was replaced by the *MUC16* 1040 fragment. HSV-TK was inserted in the E3 region. (**B**). Ad5/MUC16-1040/TK-EGFP was packaged using HEK293 cells and assessed for viral plague formation and EGFP expression, scale bar: 200 μm, magnification: 40×. (**C**). Oncolysis (CPE) of HeLa cells, scale bar: 200 μm, magnification: 40×. (**D**). Oncolysis (CPE) of OVCAR3 cells, scale bar: 50 μm, magnification: 200×. (**E**). GCV-induced cell death in the virus-infected mouse ovarian cancer line ID8 cells, *** *p* = 0.0007 vs. Ctrl, *n* = 3.

**Figure 3 cancers-13-04265-f003:**
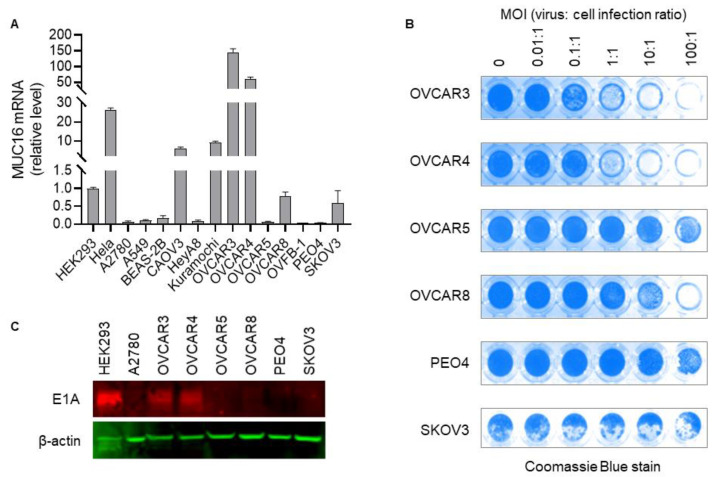
Replication of Ad5/MUC16-1040/TK-EGFP is dependent on CA-125 expression in a panel of selected ovarian cancer cell lines. (**A**). Relative *MUC16* mRNA expression in selected cell lines. (**B**). Oncolysis induced by Ad5/MUC16-1040/TK-EGFP in selected human ovarian cancer cell lines. Coomassie blue stain assay: Live cells attached to plate culture surface were stained blue. Dead cells floated and were washed out, leaving the culture surface blank and devoid of stain. Intensity of blue indicates cell density. Punctuated white areas in the SKOV3 cells were non-specific detachment of cells. MOI: multiplicity of infection, virus:cell infection ratio. (**C**). E1A expression in selected ovarian cancer cell lines infected with Ad5/MUC16-1040/TK-EGFP.

**Figure 4 cancers-13-04265-f004:**
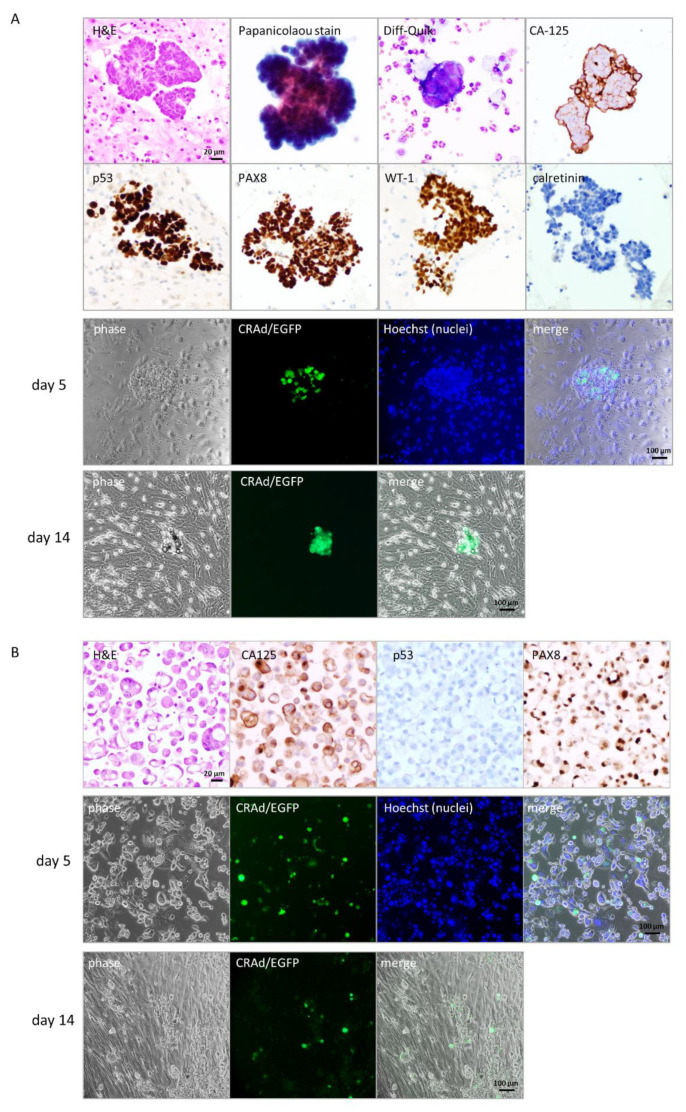
Ad5/MUC16-1040/TK-EGFP replicates in primary ovarian cancer cells. (**A**). Case 1 from malignant pleural effusion. Upper panel: Confirmation of high-grade serous ovarian cancer with expression of CA-125, PAX8, WT-1, and p53 (mutated pattern with abnormal strong diffuse expression), negative for calretinin; lower panel: Ad5/MUC16-1040/TK-EGFP replicates in primary ovarian cancer cells indicated by expression of EGFP signal, scale bar: 20 μm, magnification: 400×. (**B**). Case 2 from ascites. Upper panel: Confirmation of high-grade serous ovarian cancer with expression of CA-125 and PAX8, negative for p53 (null pattern); lower panel: predominant cancer cells with EGFP expression, indicating virus replication, scale bar: 100 μm, magnification: 100×.

**Figure 5 cancers-13-04265-f005:**
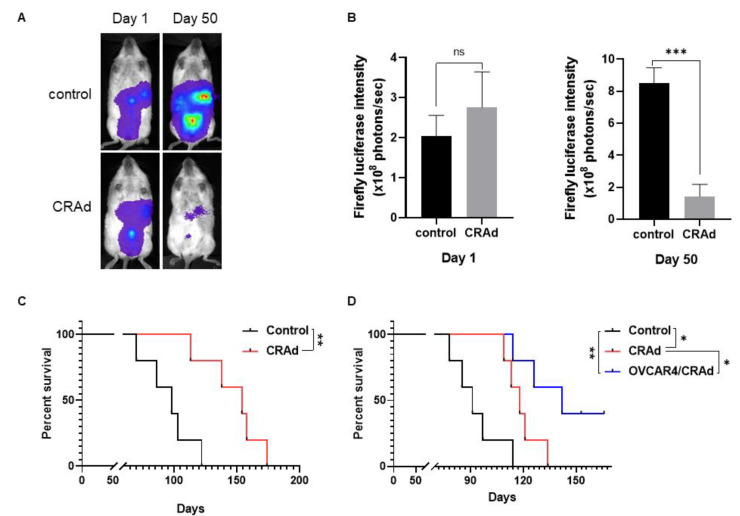
Oncolytic activity in ovarian cancer xenograft models using immunodeficient NSG mice. (**A**). Representative mouse in each group of the NSG/Kuramochi model imaged on day 1 and day 50. (**B**). Firefly luciferase intensity of NSG/Kuramochi groups on day 1 and day 50. *** *p* = 0.0004, CRAd group vs control group, ns, no statistical significance, *n* = 5. (**C**). Survival curve of NSG/Kuramochi model. ** *p* = 0.0097, *n* = 5. (**D**). Survival curve of NSG/OVCAR4 xenograft model with *n* = 5 in each group. * *p* = 0.0204, control vs. CRAd; ** *p* = 0.0028, control vs CRAd-infected cells, * *p* = 0.0326, CRAd vs. CRAd-infected cells.

## Data Availability

The data presented in this study are available in the article.

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
