# Peer review of "Targeting CA-125 Transcription by Development of a Conditionally Replicative Adenovirus for Ovarian Cancer Treatment"

_cancers, 2021, doi:10.3390/cancers13174265_

Round 1

Reviewer 1 Report

In this article, the authors describe a novel way of targeting ovarian cancer i.e. by utilizing the selective activation of MUC16 promoter in MUC16 overexpressing celll lines as compared to non-MUC16 expressing cell lines. Such strategies might one day be useful for targeting MUC16 in cancer especially when conventional strategies have failed. There are several issues in this manuscript that needs to be corrected that I have listed below.

(a) Does the adenoviral construct gets activated in mouse muc16 expressing cells? If so, does it indicate functional redundancy between human and mouse muc16 promoters?

(b) The authors have only tested cancer cell lines in their ability to activate the construct in a MUC16 expression specific way. Will the construct get activated in normal tissues where MUC16 is usually expressed for example on the cervical epithelium?

(C) Calling the 1040 bp sequence as MUC16 promoter is erroneous since the promoter has still not been properly defined. They my state that the 1040 bp sequence demostrated promoter like activity.

(D) If you knock down MUC16 in MUC16 high cell lines like OVCAR-3, will the vector still work in killing MUC16 expressing cells? The authors should perform this experiment to rule out the phenomenon.

(D) The authors should discuss the advantages of this technology for targeting intracellular targets in the disussion section since this gives it a significant edge over traditional targeting technologies that predominantly target cell surface molecules

Reviewer 2 Report

Review of cancers-1310278, Yue et al “Targeting CA-125 transcription by development of a conditionally replicative adenovirus for ovarian cancer treatment”

Summary: The submission describes the development of a conditionally replicative adenovirus that targets and lyses cells that display high CA-125 expression. Low-expressing cells are not lysed by the adenovirus. Cultured cells and cells isolated from biofluids of two ovarian cancer patients are studied. The authors then demonstrate prolonged survival among ID8-injected mice that have been treated with either the adenovirus or cells infected with the virus. The authors conclude that the model of a CA125-targeting adenovirus holds promise as a therapy modality and is worthy of further investigation.

Comments: Given the need for new treatment strategies for ovarian cancer, this study is a welcome addition. The idea of developing an oncolytic agent to MUC16/CA125-expressing cells is novel and potentially effective, as it targets cells directly, bypasses the “CA125-sink” problem associated with high CA125 levels in serum. The study progresses in a logical way from reagent development through in vitro characterization to mouse models.

I have a few comments on points that the authors should address to make the manuscript clearer:

(1) The axes legends on Figure 1A should be larger; they are currently difficult to read.

(2) An assessment of statistical significance should be included in Figures 1B and 1C.

(3) Figure 3 should be reorganized so that the bars corresponding to low-, medium-, and high-expressing cell lines are grouped together. Organizing the cell lines in this will make the trends easier to see.

(4) On page 6, the authors claim “Amplified viruses released from the cancer cells upon cell lysis can infect and kill other cells.” What is the evidence supporting this claim?

(5) Figure 3 B needs to be clearer. What do the ratios mean? What do the colors (blue/clear) mean? Why does SKOV3 look different? There is a reference in the text to Figure 3D, but there is no Figure 3D. I think the authors mean Figure 3C.

(6) The labeling in Figure 4 is unclear and inconsistent. The bolded caption heading only lists plural effusion, but both plural effusion (Case 1) and ascites samples (Case 2) are shown. Notice how the Figures are referred to in the text: they are not labeled and described consistently.

(7) On page 7, the authors mention that “there are 2 mice in this group are still alive without any evidence of cancer growth so far.” Since this article is intended to enter the historical record, it is meaningless to make this statement. The authors could provide a statement that “at the time of manuscript, corresponding to XX days post-injection, two mice in the cohort remained alive” or something to that effect.

(8) On Figure 5, some indication of statistical significance should be shown.

Minor corrections include:

(9) On page 2 “shredded” should be changed to “shed.”

(10) On page 2, rather than saying “a group from China,” the authors should list either the first author or the corresponding author by name.

(11) On page 3, “spinned” should be changed to “spun.”

(12) “HeLa” should consistently be capitalized as I have shown here, not as “Hela”

(13) Reference 21 is missing some information, such as the title and the accession date.

Author Response

Given the need for new treatment strategies for ovarian cancer, this study is a welcome addition. The idea of developing an oncolytic agent to MUC16/CA125-expressing cells is novel and potentially effective, as it targets cells directly, bypasses the “CA125-sink” problem associated with high CA125 levels in serum. The study progresses in a logical way from reagent development through in vitro characterization to mouse models.
We thank Reviewer 2 for their thorough review of our manuscript and positive feedback regarding our study design and the logical flow of the experiments. 
I have a few comments on points that the authors should address to make the manuscript clearer:
(1) The axes legends on Figure 1A should be larger; they are currently difficult to read.
Thank you for bringing this to our attention. We made the Figure 1 axes legends larger, and we circulated the manuscript to administrative staff members who assure us the figure quality, resolution, and readability are now much improved.
(2) An assessment of statistical significance should be included in Figures 1B and 1C.
We apologize for this oversight. We have now noted the statistical significance in both Figures 1B and 1C.
(3) Figure 3 should be reorganized so that the bars corresponding to low-, medium-, and high-expressing cell lines are grouped together. Organizing the cell lines in this will make the trends easier to see.
We agree with the reviewer that our figures should be easy to read. In this figure, we are presenting data from a panel of cell lines, including control cells with low CA-125 (HEK293) and high CA-125 (HeLa), normal cell lines (BEAS-2 and OVFB-1), a lung cancer cell line (A549), and various ovarian cancer cell lines with different levels of CA-125. We tried to organize the cell lines as the reviewer suggested, but this gave the impression of CA-125 trending up or down. Additionally, the control cells became mixed throughout the figure and were very difficult to pinpoint, per the feedback provided by an outside fresh-eyed colleague. In the end, we revised the figure by showing HEK293 first, since it represents the cell line against which all the other cell lines are normalized (ie, HEK293 Ca-125 level is normalized as 1). HeLa is now shown second, 
because it represents the positive control, and the remaining cell lines follow in alphabetical order. We believe the revised figure is more logically organized which makes the data easier to read and understand, 
per the overarching request of the reviewer.
(4) On page 6, the authors claim “Amplified viruses released from the cancer cells upon cell lysis can infect and 
kill other cells.” What is the evidence supporting this claim?
Our statement is based on the normal adenovirus life cycle. In culture, a host cell infected with one adenovirus will produce hundreds or even thousands of viruses, which will be released to infect surrounding cells and cause CPE that are easily visualized under a microscope. Our virus is engineered to express EGFP, 
which we usually titer in a 96-well plate utilizing serial dilution. In wells with a high concentration of virus, 
every cell will become green overnight. In wells with only a few viruses, we will see scattered green cells, 
but after a few days, we will see clusters of green cells numbering in the hundreds. After a few more days, 
every cell in the well will become green and die. 
(5) Figure 3 B needs to be clearer. What do the ratios mean? What do the colors (blue/clear) mean? Why does 
SKOV3 look different? There is a reference in the text to Figure 3D, but there is no Figure 3D. I think the authors mean Figure 3C.
Thank you for bringing these issues to our attention. To address the reviewer’s comment, we added “MOI 
(virus:cell infection ratio)” on the top of Figure 3B, and “Coomassie Blue stain” at the bottom of Figure 3B. 
We also added the following explanation to the figure legend: “Dead cells floated and were washed out, 
leaving the culture surface blank and devoid of stain. Intensity of blue indicates cell density. Punctuated white areas in the SKOV3 cells were a non-specific detachment of cells. MOI: multiplicity of infection, virus:cell 
infection ratio.” Lastly, we changed Figure 3D to Figure 3C in the text.
(6) The labeling in Figure 4 is unclear and inconsistent. The bolded caption heading only lists plural effusion, 
but both plural effusion (Case 1) and ascites samples (Case 2) are shown. Notice how the Figures are referred to in the text: they are not labeled and described consistently.
We apologize for any lack of clarity or inconsistencies. In the revised manuscript, the figure title and content were edited and brought into alignment with the photomicrographs shown and their description in the results section. This section was reviewed by a colleague with pathology training and fresh eyes, who found the revised description to be clear and accurate.
(7) On page 7, the authors mention that “there are 2 mice in this group are still alive without any evidence of 
cancer growth so far.” Since this article is intended to enter the historical record, it is meaningless to make this statement. The authors could provide a statement that “at the time of manuscript, corresponding to XX days post-injection, two mice in the cohort remained alive” or something to that effect.
We agree with the reviewer and appreciate the thoughtful suggestion. We have amended our previous statement, and it now reads: “At the time of manuscript submission, corresponding to 166 days post injection, two mice in the cohort remained alive without any evidence of cancer growth.”
(8) On Figure 5, some indication of statistical significance should be shown.
We apologize for this oversight. We have now noted the statistical significance on Figure 5B and have added 
the appropriate notation to the figure legend for 5B, 5C, and 5D.
(9) On page 2 “shredded” should be changed to “shed.”
Thank you! We have changed “shredded” to “shed” and have done a thorough review of the entire  manuscript to eliminate any other mistakes like this one. 
(10) On page 2, rather than saying “a group from China,” the authors should list either the first author or the corresponding author by name.
This is a great point. We revised this sentence, adding a reference to the author (M.X. Zhang et al) and removing the reference to their nationality (China). This sentence is now parallel or consistent with another study we reference, where the first author was mentioned, and the country of origin was not.
(11) On page 3, “spinned” should be changed to “spun.”
Thank you! We have changed “spinned” to “spun” and have done a thorough review of the entire manuscript to eliminate any other mistakes like this one. 
(12) “HeLa” should consistently be capitalized as I have shown here, not as “Hela”
We agree and apologize for this mistake. We did a search for “Hela” and all instances were replaced with “HeLa” in the revised manuscript.
(13) Reference 21 is missing some information, such as the title and the accession date.
Thank you for calling this to our attention. As suggested, we added additional information, and reference 21 
now reads: “RNA expression overview from The Human Protein Atlas. Available online: https://www.proteinatlas.org/ENSG00000181143-MUC16/cell